META-RESEARCH ARTICLE

# Leveraging vibration of effects analysis for robust discovery in observational biomedical data science

Braden T. Tierney[1,2,3,4], Elizabeth Anderson[1], Yingxuan Tan[1], Kajal Claypool[1], Sivateja Tangirala[1,5], Aleksandar D. Kostic[2,3,4], Arjun K. Manrai[1,6], Chirag J. Patel[1]*

**1** Department of Biomedical Informatics, Harvard Medical School, Boston, Massachusetts, United States of America, **2** Section on Pathophysiology and Molecular Pharmacology, Joslin Diabetes Center, Boston, Massachusetts, United States of America, **3** Section on Islet Cell and Regenerative Biology, Joslin Diabetes Center, Boston, Massachusetts, United States of America, **4** Department of Microbiology and Immunobiology, Harvard Medical School, Boston, Massachusetts, United States of America, **5** Department of Statistics and Data Science, Cornell University, Ithaca, New York, United States of America, **6** Computational Health Informatics Program, Boston Children's Hospital, Boston, Massachusetts, United States of America

* chirag_patel@hms.harvard.edu

**Data Availability Statement:** The software developed for this project, including scripts used to build the NHANES datasets, is available at https://github.com/chiragjp/quantvoe. The output of the

## Abstract

Hypothesis generation in observational, biomedical data science often starts with computing an association or identifying the statistical relationship between a dependent and an independent variable. However, the outcome of this process depends fundamentally on modeling strategy, with differing strategies generating what can be called "vibration of effects" (VoE). VoE is defined by variation in associations that often lead to contradictory results. Here, we present a computational tool capable of modeling VoE in biomedical data by fitting millions of different models and comparing their output. We execute a VoE analysis on a series of widely reported associations (e.g., carrot intake associated with eyesight) with an extended additional focus on lifestyle exposures (e.g., physical activity) and components of the Framingham Risk Score for cardiovascular health (e.g., blood pressure). We leveraged our tool for potential confounder identification, investigating what adjusting variables are responsible for conflicting models. We propose modeling VoE as a critical step in navigating discovery in observational data, discerning robust associations, and cataloging adjusting variables that impact model output.

## Introduction

Observational data science is often akin to sailing without a map. We navigate, at times blindly, through an infinite array of variables—some that were immeasurable until recently—to identify those that are potentially associated with an outcome (e.g., disease). This hypothesis-generating process, at large scale, is "discovery," and it stands in contrast to approaches underpinned by theory or a prior hypothesis (e.g., randomized clinical trials) that, for the most part, predate the observational "big data" revolution of the 21st century. Most scientific disciplines—and many eye-catching results discussed in media outlets—rely on nonrandomized/

quantvoe analysis, which can be used in conjunction with https://github.com/chiragjp/quantvoe/blob/main/manuscript/voe_pipeline_figs.Rmd to generate the figures in this manuscript, has been deposited at https://figshare.com/projects/quantvoe_manuscript_analysis_files/120969. The UK Biobank data is available upon request from https://www.ukbiobank.ac.uk/. The UK Biobank project number for this project was 22881.

**Funding:** This research is supported by National Institute of Allergy and Infectious Disease (R01AI127250, https://www.niaid.nih.gov) and National Science Foundation (1636870, https://nsf.gov) to C.J.P. and B.T.T. This research is also supported by the American Diabetes Association (ADA) Pathway to Stop Diabetes Initiator Award (#1-17-INI-13, https://professional.diabetes.org/research-grants), Smith Family Foundation Award (https://rssff.org) for Excellence in Biomedical Research to A.D.K. This research is also supported by NIH NIEHS R01ES032470 to CJP and AKM. The funders had no role in study design, data collection and analysis, decision to publish, or preparation of the manuscript.

**Competing interests:** We have read the journal's policy and the authors of the manuscript have the following competing interests: ADK is a co-founder of FitBiomics, Inc. and a member of their Scientific Advisory Board. B.T.T. consults for Seed Health on microbiome study design and analysis.

**Abbreviations:** AIC, Akaike information criterion; BMI, body mass index; COVID-19, Coronavirus Disease 2019; FDR, false discovery rate; GWAS, genome-wide association study; INT, inverse normal transformation; JE, Janus effect; LDL, low-density lipoprotein; NHANES, National Health and Nutrition Examination Survey; RT-PCR, reverse transcriptase polymerase chain reaction; VoE, vibration of effects.

noninterventional observational data in some capacity. Whether it be eyeglass usage and COVID-19 infections [1], milk and breast cancer [2], the human microbiome and any number of diseases [3], or red meat and heart disease [4], these conclusions are based on correlations from observational studies of varying size.

Charting a course through the process of analyzing observational data (Fig 1) is not always straightforward, however. Any decision along the analytic pipeline—from data collection to statistical analysis—can introduce uncertainty (e.g., false positives), obscuring underlying biology [5–7]. As a result, there is a large need for methods to aid in navigating the maelstrom of the analytic process. Here, we present a tool, "quantvoe," for mitigating one of many potential biases: arbitrary model specification to address potential confounding. Broadly speaking, quantvoe operationalizes the identification of robust (i.e., high-confidence) versus nonrobust (i.e., low-confidence) associations, enabling greater long-term model reproducibility and interpretability. For example, if our tool had existed in the past year, we could have understood one aspect of the confusing and contradictory claims regarding COVID-19 positivity and vitamin D levels [8].

Quantvoe identifies confounding in associations by a form of sensitivity analysis: modeling "vibration of effects," (VoE), which is defined as how model output changes as a function of specification, specifically the distribution of the size, direction, and statistical significance of the association between 2 features (Fig 1B–1D) [5–7]. It is capable of identifying highly nonrobust associations with "Janus effects" (JEs), where model specification yields consistently results, reporting both positive and negative associations. In practice, JEs can be used to understand if an association between 2 variables is consistent regardless of model specification. For a full list of definitions relevant to quantvoe, please see S1 Table.

The task of correlating an "independent variable X" with a "dependent variable Y" lies at the heart of many observational studies. Confounding variables are defined as being correlated to both X and Y. A priori "adjustment," or inclusion/stratification of hypothesized and measured confounding variables in a model, may address confounding bias; however, there is an infinite sea of modeling options. Without underlying theory, there is no consensus on what variables should be considered in a model [6]. However, arbitrary choices in modeling strategies has leading some to conclude that most published research findings are false, or, at least, not reproducible [7]. Different adjustment or modeling strategies may lead to contradicting results—an "X," (e.g., vitamin D), being either positively or negatively associated with a "Y" (e.g., COVID-19 positivity) depending on what covariates a researcher adjusts for in a given model. In other words, is vitamin D really associated with COVID-19 positivity [8], or are there some other characteristics of individuals that have high vitamin D (e.g., age, exercise, diet) that can make them appear at higher or lower risk for COVID-19?

Modeling VoE can be thought of as a specific type of "multiverse analysis" designed for a "discovery" scenario in biomedical sciences, where there are many possible X variables to be correlated with a Y variable, such as metagenomic-phenotype association studies or exposome-wide association study [9,10]. Multiverse analyses, very broadly speaking, describe how any analytic choice (e.g., data preparation, model assumptions, model type) can change findings, and there are many different forms of them [11–13]. But few of the possible types of multiverse analyses have been fully automated and many require manual configuration. Additionally, some of these tools, such as specification curve analysis [14,15], rely on users "enumerating all data analytic decisions necessary to map a hypothesis onto a set of possible associations," a step often not feasible for complex, discovery-based questions and analyses, such as genomics. VoE expands on multiverse analysis, fully automating the exploration of model choice. For example, we have recently used VoE to prioritize >1 million correlations in metagenomic-phenotype associations [16].

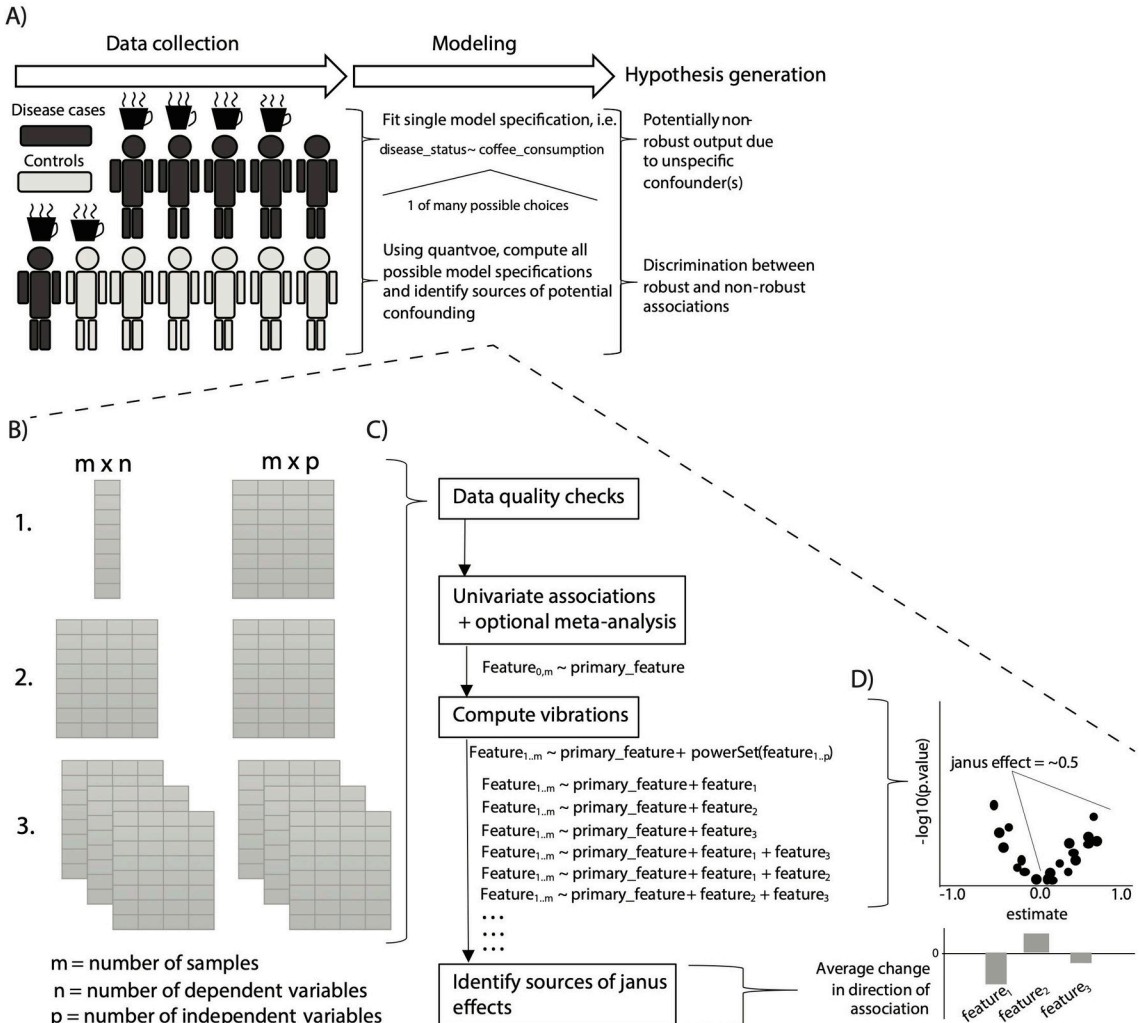

**Fig 1. Overview of the algorithmic approach and its place in the hypothesis generation toolkit.** (A) The process/paradigm of "data-driven" discovery. First, high-dimensional data are collected on a set of individuals with a given phenotype (i.e., a disease) as well as controls (individuals without the phenotype). The researcher selects a modeling strategy, most often just one, and computes associations between the phenotype of interest (a "Y") and a particular feature (an "X," like coffee in the example). The resultant associations yield associational "hypotheses" regarding the relationship between the X and the Y, but only fitting one model specification can yield nonrobust results. Quantvoe replaces this middle step of choosing one model, instead attempting to fit up to every possible model specification, thereby charting a course to robust hypothesis generation. (B) Quantvoe takes 3 types of input data, all in the form of pairs of data frames (tables), either at the command line or in an interactive R session: (1) a single dependent variable and multiple independent variables; (2) multiple dependent variables, or (3) multiple datasets. (C) There are 4 main steps—checking the input data, computing initial univariate associations, computing vibrations across possible adjusters, and quantifying how adjuster presence/absence correlates to changes in the primary association of interest. (D) Following computing VoE, we evaluate the results by measuring Janus effect (the fraction of associations greater than 0) and estimating the impact of different adjusters on the change in correlation size.

No standardized software package exists to systematically model VoE in massive datasets with hundreds of potential adjusting variables. Other work and software packages for modeling sensitivity under the umbrella of multiverse analysis is additionally focused less on modeling strategy and more on alternative aspects of data analysis, such as data collection [17]. We propose that this gap is limiting the ability of researchers to navigate observational data, identifying robust associations and potential confounders. Indeed, in other disciplines (e.g., genetics), the development of analogous software (e.g., PLINK [18]) democratized genetic

epidemiology and meta-analysis (e.g., via genome-wide association studies [GWASs]). Finally, there is also a need also missing from these other approaches to not just quantify VoE demonstrate how results change as a function of approach, but specifically show why model output may vary why they do: what adjusters confound associations the most and therefore should be considered in future studies.

To address these gaps, we developed quantvoe to efficiently search among all (potentially thousands) of covariates, automatically identifying what measured adjusting variables drive inconsistency in associations between X and Y. Simply put, for a given dataset, our package fits up to every possible model using every covariate present and analyzing variation in the output (Fig 1). This analysis yields lists of candidate variables, some of which potentially could be considered as adjusting variables in later analyses. We specifically analyze the degree to which variables potentially impact association sizes.

We applied our package to 2 use cases to demonstrate its utility for addressing the critical step of identifying robust versus nonrobust associations in biomedical research. First, we used 2 of the largest, general observational cohorts in existence (The volunteer-based UK Biobank [UKB] and the survey-based National Health and Nutrition Examination Survey [NHANES]) to query 5 different associations. We selected these cohorts on the bases of their vast array of potential confounding variables (e.g., laboratory, dietary, and lifestyle covariates) and sample size, each containing >50,000 individuals. Second, in NHANES, we explored VoE in-depth in the association between a set of dietary and lifestyle exposures and indicators of risk for cardiovascular disease and medical intervention.

## Results

### Vibration of effects pervades popularized and reported associations

We probed the associations between calcium intake and femur density, carrot intake and eyesight, COVID-19 positivity and vitamin D levels, poverty level and blood glucose, and lisinopril usage and systolic blood pressure. Lisinopril, which is used to reduce blood pressure, is the most commonly prescribed drug in the NHANES 2011 to 2018 prescription data. Socioeconomic status is often reported as a risk factor for type 2 diabetes (high blood glucose), and vitamin D has been both positively and negatively associated with COVID-19 positivity in conflicting studies [8,19–21]. Finally, calcium and carrot intake pervade the public zeitgeist as causal for increasing bone strength and vision, respectively [22–25].

Fitting 10,000 vibrations (different model specifications) per association, we identified varying levels of VoE in the 5 potential relationships described above (Fig 2A–2E, S2 Table). In addition to reporting JEs, we report the median, 1%, and 99% quantiles of association sizes across all models. SARS-COV-19 (COVID) positivity and vitamin D (Fig 2E) had the most substantial JE (0.55, i.e., 55% of models show a positive relationship, 45% a negative one), followed by systolic blood pressure and lisinopril (Fig 2D) (JE = 0.46), though the latter had fewer nominally significant outputs than the other associations tested. The association between poverty index and blood glucose had a JE of 0.09 and was overall negative (i.e., decreased wealth, increased blood glucose).

We found that moderate changes to model adjustment yielded contradictory results. For example, adjusting the association between femur density and calcium intake, for phosphorus intake versus vitamin B12 intake yielded opposite sign yet statistically significant results (Fig 2A). Similarly, alternative dietary variables (e.g., egg, fruit, and vegetable intake) appeared to correlate to sign-switching in the relationship between carrot intake and vision, which was our most robust finding (though it still had a moderate JE = 0.06). Incidentally, however, on average, increased carrot intake was associated with worse vision (a median negative association

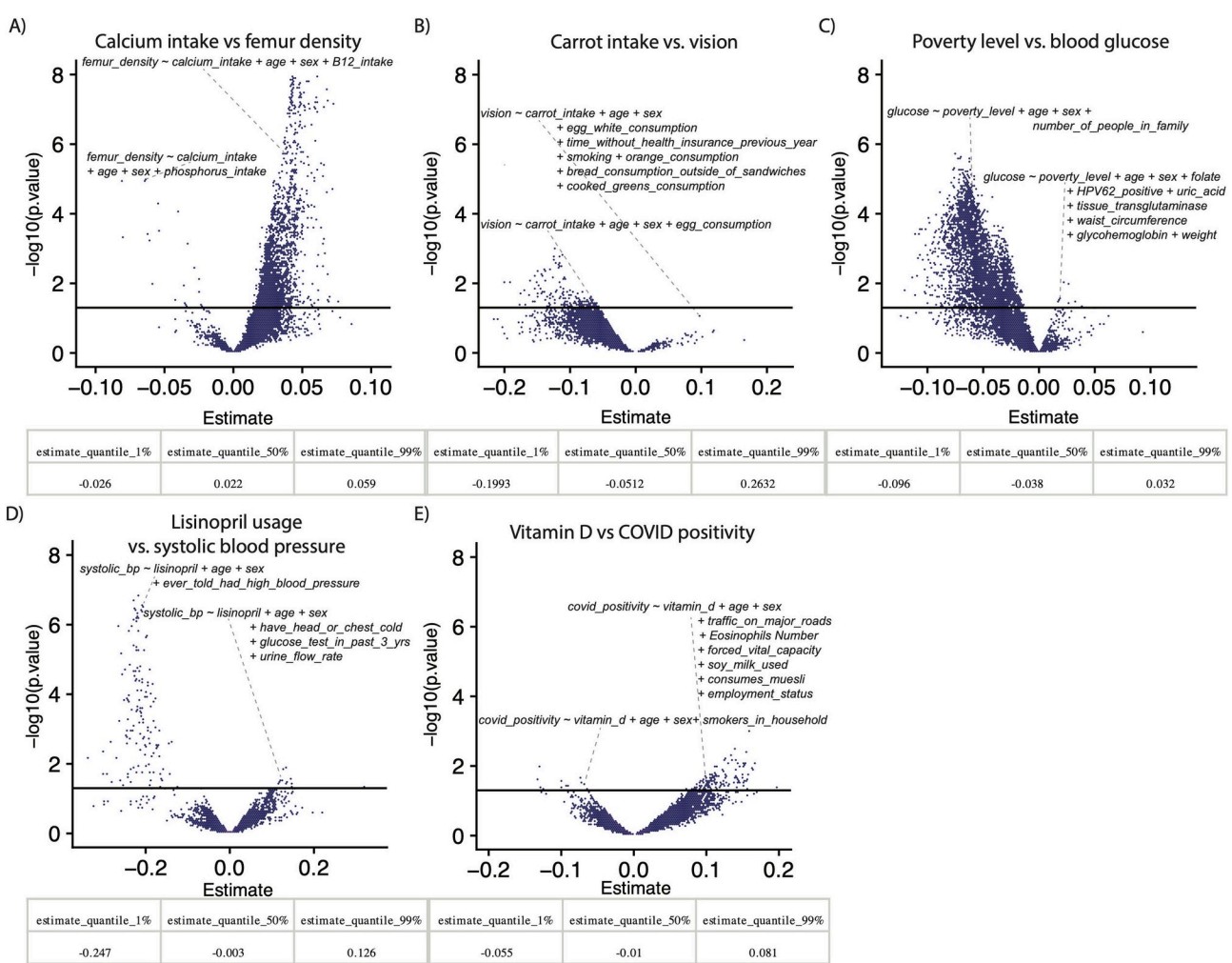

**Fig 2. Examples of VoE for prominent associations.** Each point in the density plots represent at least one model. x-Axes are estimate size for beta-coefficients of interest (e.g., for panel A, the coefficient on the calcium intake variable). Quantiles show the range of estimate sizes for each above relationship. The y-axis is the −log10(p.value) of that association. The solid line is nominal (*p* < 0.05) significance. Data underlying these plots are available at https://figshare.com/account/home#/projects/120969 and S2 Table. VoE, vibration of effects.

size of −0.05). Adjusting the COVID-19 versus vitamin D associations for the presence of smokers in the patient's household in one case yielded a nominally significant negative association, whereas other models, when adjusted for a range of features like dietary choices, yielded a positive and nominally significant result.

## Quantifying vibration of effects in common indicators of cardiac health

We next leveraged our pipeline to model VoE for a second use case—querying a range of indicators that have been reported as relevant to aspects of cardiovascular health and are component variables of the Framingham Risk Score [26–28]. Specifically, our dependent variables were systolic blood pressure, body mass index (BMI), low-density lipoprotein (LDL) cholesterol, and total cholesterol. We tested the association between these phenotypes and exposures that had been reported as associated in the observational literature with cardiovascular health, including total caloric intake, fiber intake, alcohol intake, sugar intake, fat intake, caffeine

intake, physical activity, smoking, family history of coronary artery disease, and, once again, BMI [26,29–35].

We identified VoE for the majority of these 39 relationships (Fig 3, S2 Table), with 26/39 (66.7%) of tested relationships highly dependent on model specification and demonstrating JEs. Only one association (between systolic blood pressure and alcohol usage) was devoid of conflicting results, and 30/39 (77.0%) had at least one example of statistically significant ($p < 0.05$) and opposite sign associations for the variable of interest. The least robust associations were between LDL and sugar intake (JE = 0.53), systolic blood pressure and total fat (JE = 0.48), and BMI and caffeine intake (JE = 0.55). The most robust negative associations were systolic blood pressure and caffeine intake (JE = 0.001) as well as systolic blood pressure and fiber intake (JE = 0.002). The most robust positive associations were systolic blood pressure and alcohol intake (JE = 1.0), total cholesterol and caffeine intake (JE = 1.0), total cholesterol and alcohol intake (JE = 1.0), systolic blood pressure and BMI (JE = 1.0), and LDL and caffeine intake (JE = 1.0). With the exception of its association with caffeine intake, every relationship tested with LDL was nonrobust.

Having discovered substantial model specification–dependent VoE among established associations with cardiovascular health, we next sought to identify which adjusters were driving the majority of JEs. As part of its built-in function, our software measures how the presence or absence of different variables correlates with increases or decreases in the absolute value of association size for particularly nonrobust associations (see Materials and methods, Eq 11).

We report the impact on VoE of (1) established correlates of heart disease risk as well (education and race/ethnicity) as (2) additional adjusters illuminated by our study (Fig 4). We were able to identify situations where the presence or absence of an adjuster in a model correlated heavily to the direction of association for the independent and dependent variables of interest. Alternatively, dietary intake (e.g., magnesium and moisture) tended to have strong effects on the direction of association between fiber intake and total cholesterol. (Fig 4A and 4B). The association between smoking and systolic blood pressure (Fig 4C and 4D) was sensitive to adjustment to socioeconomic variables (e.g., income). Notably, models adjusted for education tended to have negative directionality, whereas models adjusted for weight tended to have positive directionality. Adjusting for other dietary and socioeconomic features, such as insurance coverage, also played a major role in the associations between BMI and caloric intake (Fig 4E and 4F) and LDL and sugar intake (Fig 4G and 4H). For the latter, modules adjusted for carbohydrate intake tended to yield a positive association, whereas models adjusted for if an individual had received an oral glucose tolerance were negative. We additionally report model validation plots for this analysis (e.g., residuals) in S1 Fig.

We found in many cases that the same adjusting variables affected associations for multiple dependent features, such as in the case of carbohydrate intake, risk for diabetes, and weight. Conversely, a number of dominant adjusters were unique to particular dependent features, like folate and total cholesterol/fiber, various fatty acids and BMI/caloric intake, and hemoglobin and blood pressure/smoking.

## Toward optimally modeling of vibration of effects

Modeling VoE is inherently a computationally demanding task due to the number of models that must be estimated. As such, we estimated the minimum number of models fit to accurately measure association robustness or associations of interest that we would later probe in detail. We fit a total of 13,047,200 models. Runtime scaled exponentially, whereas RAM usage was relatively unchanged (Fig 5A and 5B, S2 Fig), with a range of 0.38 to 6.5 GB depending on the number of vibrations and specific association.

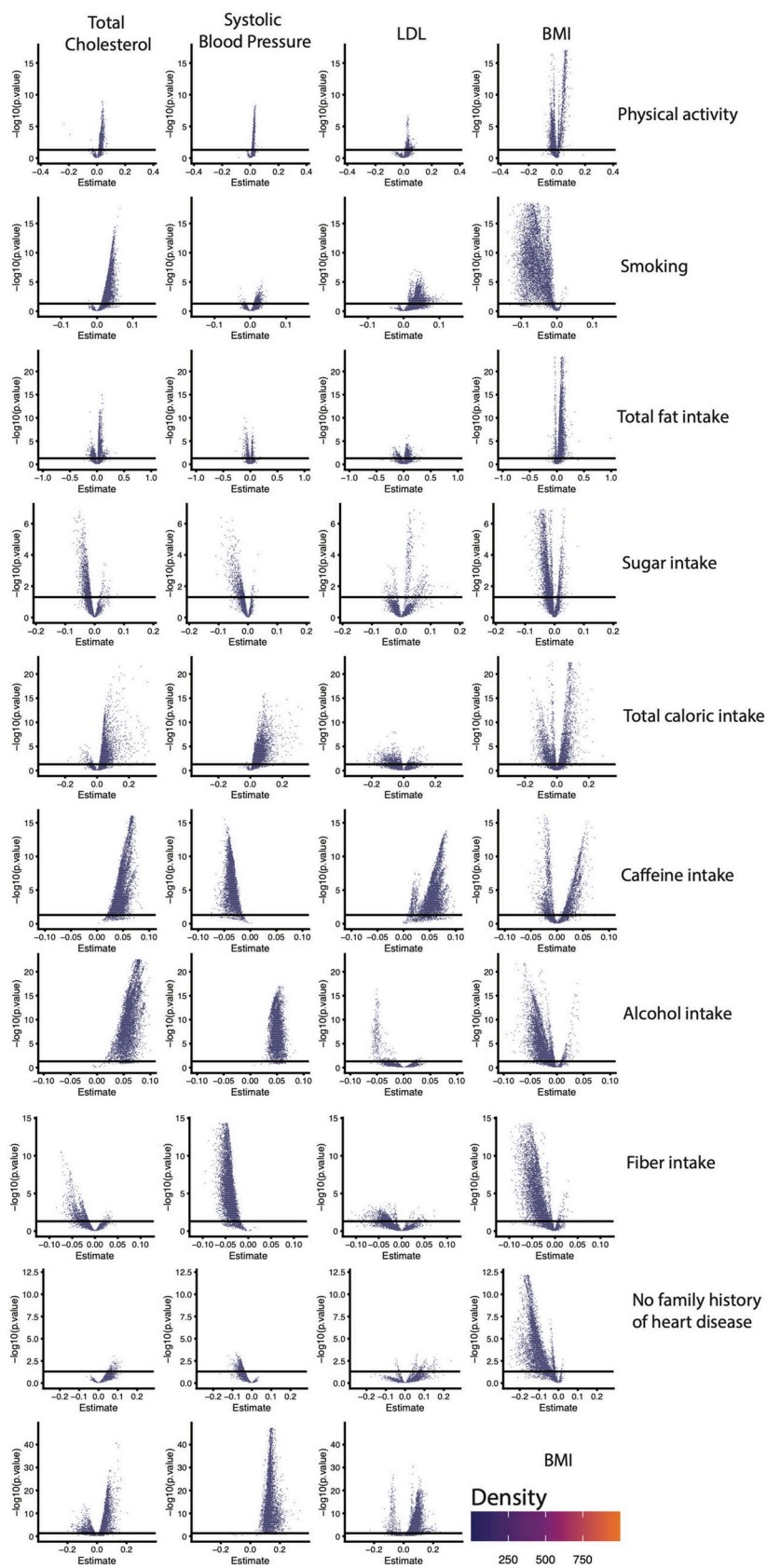

**Fig 3. VoE output for 10 exposures and their relationship with common indicators of cardiovascular health in the form of density plots.** Each point represents output from different model specifications. The x-axis is the *beta*-coefficient on the primary independent variable of interest (e.g., physical activity), and the y-axis the −log10(p.value) of that association. The solid line is nominal ($p < 0.05$) significance. Data underlying these plots are available at https://figshare.com/account/home#/projects/120969 and S2 Table. BMI, body mass index; LDL, low-density lipoprotein; VoE, vibration of effects.

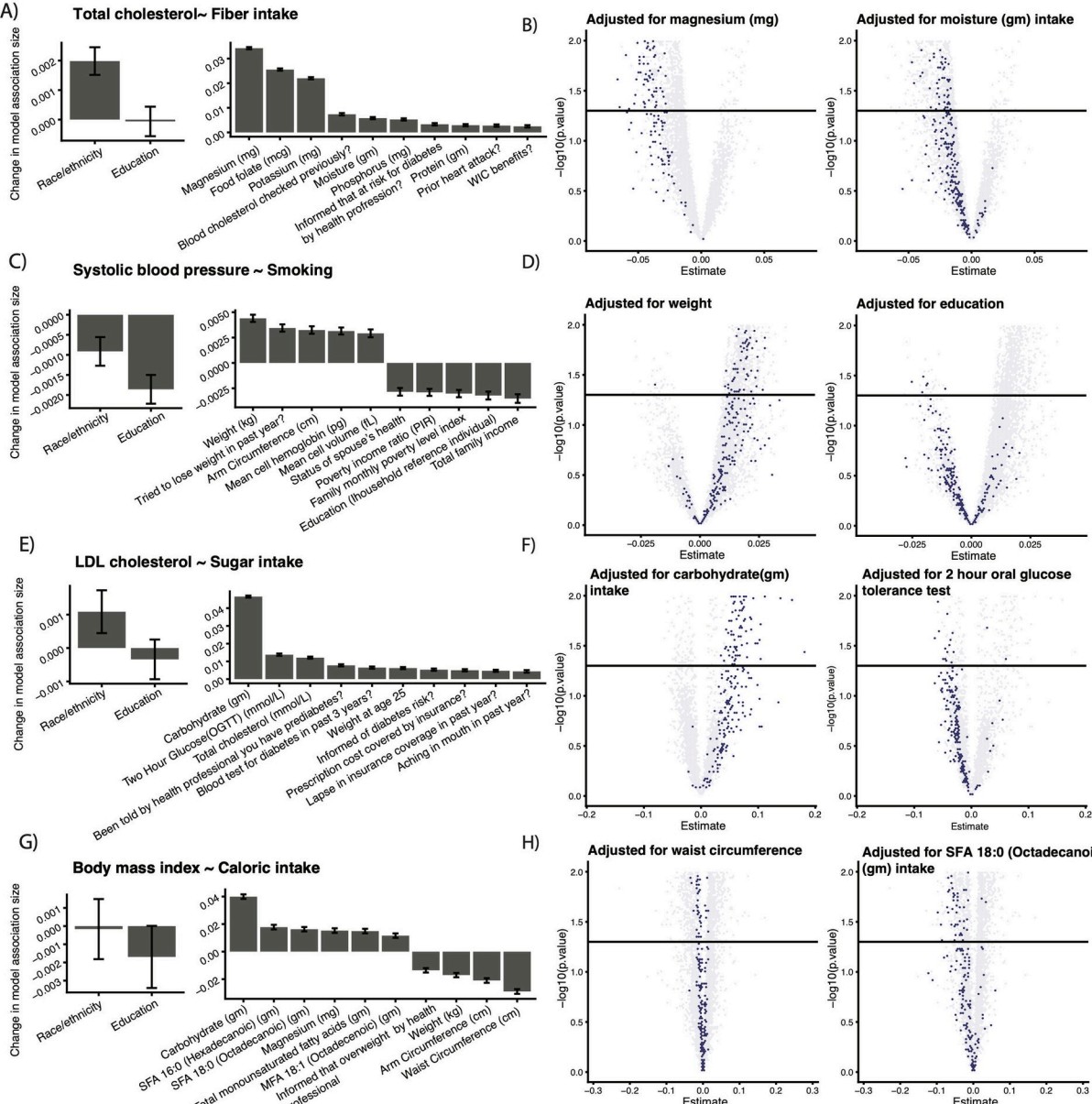

**Fig 4. Examples of adjusters that appear to drive VoE for highly confounded associations.** Rows: associations with particularly high JEs for each dependent variable. Left column: established variables known to confound cardiovascular risk factors. Middle column: top 10 adjusters (not found in the left most column) whose presence is correlated to changes in the associations reported on the rows. Right columns: vibration plots from Fig 3 (rescaled on the axes) colored by if the adjuster in the plot title was present in models represented by a given point. The solid line is nominal ($p < 0.05$) significance. Data underlying these plots are available at https://figshare.com/account/home#/projects/120969 and S2 Table. JE, Janus effect; LDL, low-density lipoprotein; MFA, Monounsaturated Fatty Acids; OGTT, oral glucose tolerance test; PIR, poverty income ratio; SFA, Saturated Fatty Acids; VoE, vibration of effects; WIC, Women, Infants, Children.

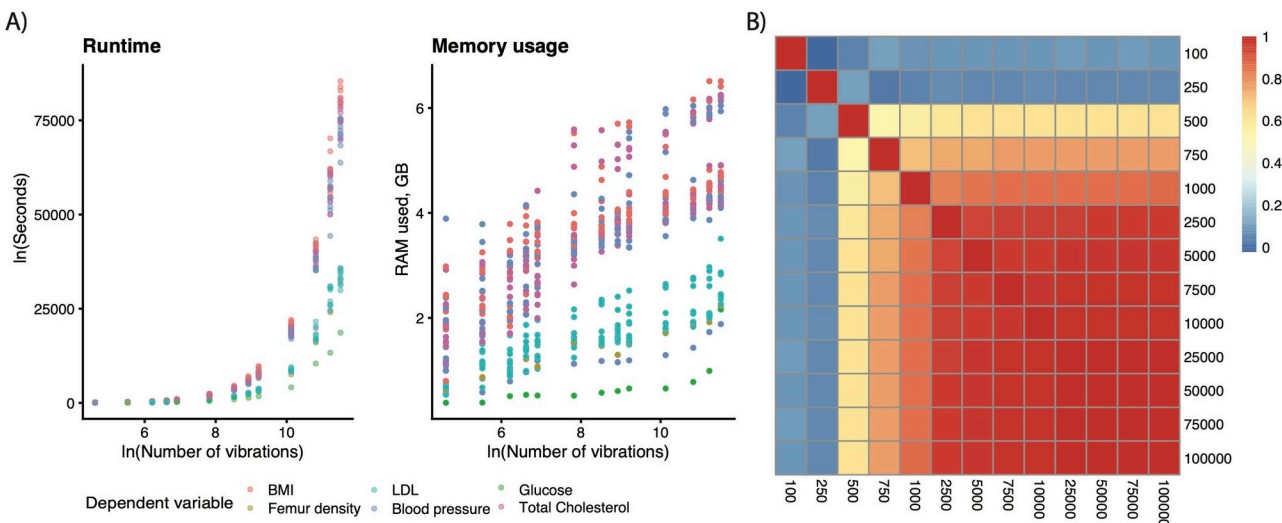

**Fig 5. Software performance statistics.** (A) Runtime for different numbers of vibrations, (B) memory usage for different numbers of vibrations, and (C) consistency of results across the number of vibration associations (n). Heatmap color depicts the correlations between the coefficients in the nonmixed effects version of Eq (10). Data underlying these plots are available at https://figshare.com/account/home#/projects/120969. BMI, body mass index; LDL, low-density lipoprotein.

We measured the impact of the presence or absence of different adjusting variables on the association between our primary variables and dependent variables. The output of this analysis is a weight measuring the impact a given adjusting variable has on change in association size—we computed the correlation between these weights at differing numbers of models fit (e.g., 100 versus 1,000 versus 10,000). At 10,000 vibrations (which ran for an average of 1.8 hours and used 3.4 GB of RAM) per dependent variable, the correlation with all higher-numbers-of-vibration tests was on average >0.98 in all cases, indicating consistency in the effects of different adjusters regardless of sampling size above that point (Fig 5C). Given our other parameters (max variables per model = 20), this worked out to each of the approximately 300 adjusting variables (the exact number varying depending on the association in question) being present an average of 398 +/− 178 models when running 10,000 vibrations.

## Discussion

The novelty of our tool—quantvoe—lies in its ability to bridge the gap between data and robust associations in observational and biomedical studies. By using quantvoe, researchers can identify which associations are robust (or not) and why this is the case prior to publishing them. Indeed, many "popular" associations—like the association between egg consumption and mortality [36–38]. The association between wine consumption and cardiovascular health [39,40] and even the results of famous "marshmallow experiment" as an indicator of a child's self-control and success later in life [41–44] have conflicting reports of directionality that, we hypothesize, could potentially be attenuated by multiverse analyses such as probing the space of VoE possible for those variables.

Given our result showing the immense variation in output even in well-established correlates of cardiovascular health (e.g., exercise and BMI) and other relationships (e.g., vitamin D and COVID-19 positivity), we clearly need to be transparent when anchoring claims based on associations, especially when hypotheses about the *X* and *Y* are not prespecified or when arising from datasets with a large number of variables measured. In the predictive setting, and even in the case where a researcher prefers not to use a linear model for their final association

or prediction (and would rather compute a final predictive accuracy statistic) modeling VoE quickly on the important variables can still be useful to quantify the robustness of predictions [45].

Our approach, however, is not without caveats. First, our specific implementation makes use of generalized linear models, and there are many possible more complex associational methods that could potentially capture nonlinearities, such as pretransforming the variables (which itself could be thought of as yet another parameter to vary). Second, our definition of JE is only contingent on association size, and not statistical significance. An alternative approach would be to measure JEs only for estimates that passed a certain significance cutoff. We deliberately chose to avoid this as (1) $p$-values vary highly depending on small changes made by the researcher and significance thresholds are, at best, somewhat subjective, and (2) while approaches exist for inference after feature selection in machine learning (e.g., "post-selection inference" [46]), it is still not straightforward to estimate inferential statistics on many popular modeling approaches like random forests or similarly complex machine learning methods. It is additionally worth noting that when signs change sign, or the JE, across model specifications is neither entirely unexpected (due to the underlying correlation structure of the data) nor necessarily a negative [47] with respect to overall findings. In the examples we describe, the JE could help identify potential confounding variables.

Additionally, our analysis to identify potential confounders (Fig 4) may have some drawbacks if the models violate some of the assumptions of regression. While, on average, the examples in this manuscript did not egregiously do so (S1 Fig), there are, for example, outliers and slight nonnormality in residuals. As a result, we recommend users consider the full model output, which is provided alongside summary statistics, for their specific use case. In future versions of quantvoe, it may be prudent to consider alternative confounder analysis strategies than a simple (mixed) linear regression, like variable selection methods (e.g., an elastic net).

There are also potential drawbacks to our brute force modeling approach (i.e., fitting up to every possible model). For example, there may be variables not measured in a cohort that are critical for modeling, and given that random permutations of models are fit, these may be missing a certain unknown necessary adjusting variable. Alternatively, collider bias or reverse causality may complicate the output of the mixed effect modeling analysis in terms of distinguishing between confounding and colliding. We also do not consider metrics such as $R^2$ or Akaike information criterion (AIC), which could be used to evaluate model fit. However, full model outputs are returned as part of quantvoe, so users can analyze these metrics. Future iterations of quantvoe should automate the evaluation of models based on these and other criteria, enabling users greater clarity when interpreting our analysis.

Given these drawbacks, it is worth noting that there are other tools, like Bayesian modeling averaging [48] or regularization methods (e.g., LASSO) [49], which have been used to search the space of variables measured in a dataset that predict the $Y$ optimally; however, these stand apart from our method, which automates multiverse analysis by undertaking a systematic exploration and presentation of the association and inference between an $X$ and $Y$ as a function of all other possible models in a dataset. Furthermore, there are other methods that could be used for sensitivity analysis. A nonexhaustive list includes E-values (which measures how strong the association between potential confounding variables, the outcome, and the exposure would have to be to influence an association of interest), Monte Carlo sensitivity analysis (which attempts to correct for bias based on model parameters), or the Confidence Profile Method (which is a Bayesian approach to measure uncertainty) [50,51]. All of these methods have their strengths and limitations [52]. The key difference between these tools and VoE is that VoE explicitly models the impact of specific and measured adjusting variables or model specifications on an association size instead of using outside parameters and assumptions to

do so. That said, these methods could be used to augment quantvoe, or potentially they could be deployed in its stead in situations where the potential drawbacks of the brute force modeling approach (e.g., compute time) outweigh the benefits. We believe considering quantvoe alongside other tools will be crucial in the future, as its misuse or overuse would lead to an excessive on how it displays model robustness, which, in turn, would be its own form of bias.

Overall, we have developed software for automatically assessing VoEs to identify adjusting variables that may foul up associations, leading investigators and the public astray. This software allows researchers to automate sensitivity analyses, to navigate through large-scale association studies, fulfilling a tenet of biomedical data science: computing robust associations. We used our tool to query a range of reported associations, including those between exposures and cardiovascular risk prediction, the components of the Framingham Risk Score [28]. In the end, our methods are generalizable to any correlational study. We hope that our tool can be deployed, either prior to or in lieu of more complicated associational methods, to deepen our understanding of "rules" that govern observational data, providing a bulwark against the publication of inaccurate results and prioritizing the impact of those that are robust.

## Materials and methods

### Overview

The "quantvoe" package can be accessed from Github (https://github.com/chiragjp/quantvoe). It is implemented using the Tidyverse [53], so all data frames should be passed as "tibbles." For more information on the Tidyverse and its associated data structures (e.g., tibbles), please see https://www.tidyverse.org/. Its purpose is, for a given dataset, to identify the "analytic flexibility landscape" of a particular correlation between a "primary independent variable" (e.g., BMI) and at least one "dependent" variable (e.g., dietary fiber intake) in 3 broad steps: (1) initial associations; (2) vibrations; and (3) analytic flexibility landscape analysis (identifying potential confounding adjusters, as in Fig 5) (Fig 1C). These processes involve, respectively, fitting an initial univariate linear model for an independent primary variable of interest and each dependent primary variable, computing vibrations for each dependent variable (i.e., fitting up to $2^P$ models, where $P$ corresponds to the number of possible adjusters and generally should be under 1,000), and identifying with a separate regression analysis the contribution of different adjusters to the overall VoE. If multiple cohorts are provided, the software will run a random effects meta-analysis after the first step to compute overall $p$-values and effect sizes for the initial associations. An analyst can use any family of linear model from R's "glm" module (with the default being gaussian). Another option includes negative binomial regression for overdispersed count data. The output from each step, including the initial data, is returned as a named list. The entire pipeline can be run with one command, or its components can be deployed individually. In addition to the standard, interactive, function-based interface of the R language, we provide a command line interface to allow for straightforward package deployment on virtual machines or high-performance-compute clusters. We confirm that we followed the STROBE checklist for reporting of observational studies.

### Parameters

We provide a range of parameters (S4 Table) when running the full pipeline that allow users to customize the pipeline to their specific use case and, if needed, reduce processing. That said, only 3 parameters must be specified by the user, as the rest have defaults that will work in most cases.

If the user is using the command line implementation, then these parameters can be displayed by running "Rscript voe_command_line_deployment.R -h." The only additional

required parameter is "-o," which specifies the path to and name of the RDS file where the output should be saved.

## Input data

User input can come in 1 of 3 forms depending on the end goal (Fig 1B). In all cases, the user specifies at least 1 pair of R tibbles—one containing the dependent variable(s) of interest ($m$ X $n$, where m = number of samples and $n$ = number of dependent variables and sample identifiers), one ($m$ X $p$, where p = number of adjusters, the independent variable, and the sample IDs) containing the independent variables (the primary independent variable of interest and the adjusters to be used in vibrations). The first (left-most) column for both tibbles links the dataframes and must contain the sample identifiers, with one unique value per row. Multiple samples per individual (i.e., duplicate sample identifiers) are not permitted. The numbers on the left hand side of Fig 1B correspond to the following 3 situations:

1. One dependent variable—This is a classic cross-sectional epidemiological design. The dependent variable tibble contains 2 columns, one for the sample identifiers and one for the data.

2. Multiple dependent variables (discovery)—The user has multiple dependent variables (many thousands) of interest and is interested in discovery.

3. Multiple cohorts (meta-analysis)—The user has multiple cohorts and therefore must supply as many pairs of dependent/independent dataframes as there are cohorts. After running the initial associations for each cohort, a meta-analysis will be computed to generate overall summary statistics for the correlation(s) of interest.

As input, a user can specify any number of cohorts or dependent variables to vibrate over. If multiple cohorts are provided, the tool runs a random effects meta-analysis across the initial associations in each cohort (see Initial associations and optional meta-analysis).

## Initialization

Prior to launching the pipeline, the tool reports and logs the specific parameters of the run (see Parameters). It additionally checks the input data to confirm that the sample identifiers match and can be merged, that the number of independent and dependent tibbles match, and that the column names of the dataframes do not conflict with any global variables required for the pipeline to run.

## Initial associations and optional meta-analysis

For each of the dependent variables and for each dataset, an initial, univariate association is computed using the linear regression family selected by the user (Eq 1). Note that this equation can be modified to include X constant (baseline) adjusters—independent variables that will be present in every vibration (not depicted below).

$$\text{Dependent Variable} \sim \beta_0 + \beta_1 \text{Primary Variable} \tag{1}$$

where $\beta_1$ is the association between the *primary variable* and the *dependent variable*. Overall, this step returns the list of dependent variables that will be "vibrated" over as well as the raw association output for each one. The pipeline can remove dependent variables that are above a sparsity threshold (e.g., those that are 99% zero values) set by the user with the proportion_cutoff parameter. Any generalized linear model offered by R's *glm* command can be used, as well as negative binomial regression (from the MASS package) or survey-weighted regression

(from R's survey package). Summary statistics are calculated for each regression, and $p$-values are adjusted for multiple hypothesis testing with a method that can be selected by the user. Given that the pipeline is designed for computing VoE for a large set of dependent variables (e.g., microbiome data), only those that have $p$-values below and false discovery rate (FDR) cutoff threshold (that can be set by the user) will be selected for vibration analysis. To select all dependent variables regardless of initial significance, the FDR threshold can be set to 1; however, for massive datasets, this can lead to long computation times. Regressions for dependent variables that return warnings or errors are dropped from the pipeline.

Providing multiple datasets to the pipeline will trigger a random effects "REML" [54] meta-analysis (from which all relevant outputs are reported, including estimates of heterogeneity, including $I^2$ and Q values). In this step, the results from each of the univariate associations will be combined across each dataset to generate an overall $p$-value and estimate size. These $p$-values can be adjusted for multiple hypothesis testing (e.g., FDR) and filtered in a similar manner to the non-meta-analytic pipeline in order to identify features for the vibration analysis.

## Vibrations: Computing the distribution of association sizes due to model specification

The pipeline computes VoE for each of the features that were selected in the "initial associations" step (Fig 1B, Eq 2), using the same regression parameters (such as sampling weights, etc.). This is done by fitting a series of models containing the same primary and dependent variable as in Eq 1; however, this time adjusted by, at most, every possible combination of adjusting variables (or "adjusters").

$$\text{DependentVariable}_n \sim \beta_0 + \beta_1 \text{PrimaryVariable} + \beta_{2..n} \text{powerSet(adjusters)} \tag{2}$$

Eqs 3 to 9 demonstrate the models that would be fit in the case of 3 adjusters and 1 dependent variable:

$$\text{DependentVariable} \sim \beta_0 + \beta_1 \text{PrimaryVariable} + \beta_2 \text{adjuster}_1 \tag{3}$$

$$\text{DependentVariable} \sim \beta_0 + \beta_1 \text{PrimaryVariable} + \beta_2 \text{adjuster}_2 \tag{4}$$

$$\text{DependentVariable} \sim \beta_0 + \beta_1 \text{PrimaryVariable} + \beta_2 \text{adjuster}_3 \tag{5}$$

$$\text{DependentVariable} \sim \beta_0 + \beta_1 \text{PrimaryVariable} + \beta_2 \text{adjuster}_1 + \beta_3 \text{adjuster}_2 \tag{6}$$

$$\text{DependentVariable} \sim \beta_0 + \beta_1 \text{PrimaryVariable} + \beta_2 \text{adjuster}_1 + \beta_3 \text{adjuster}_3 \tag{7}$$

$$\text{DependentVariable} \sim \beta_0 + \beta_1 \text{PrimaryVariable} + \beta_2 \text{adjuster}_2 + \beta_3 \text{adjuster}_3 \tag{8}$$

$$\text{DependentVariable} \sim \beta_0 + \beta_1 \text{PrimaryVariable} + \beta_2 \text{adjuster}_1 + \beta_3 \text{adjuster}_2 + \beta_4 \text{adjuster}_3 \tag{9}$$

The coefficient of interest is that on the primary variable ("PrimaryVariable"), $\beta_1$. It indicates the association between the primary and dependent variables. When we refer to "VoE," we are considering the distribution of the size, direction, and statistical significance of this value for all models considered (e.g., Eqs 3 to 9).

The maximum number models that can be fit is $S^*N^*2^P$, where $P$ is the number of adjusters provided by the user, $N$ is the number of dependent variables provided, and $S$ is the number of datasets provided. The number of models fit can be lowered by adjusting the

max_vars_in_model and/or max_vibration_num parameters, which, respectively, reduce the number of variables that can be in a given model and reduce the overall number of models fit per feature. Setting either of these will force the pipeline to randomly select combinations of variables (with no more in a single set than the max_vars_in_model parameter allows), and then vibrating over those sets. The functions additionally drop any independent categorical adjusters that lack multiple levels. The vibration process also can be sped up and distributed across multiple CPUs using the "cores" argument. This component of the pipeline returns each of the results of each regression for each dependent feature in a large, nested tibble.

### Identifying drivers of association biases

By default, the pipeline will attempt to identify the major drivers of VoE for the user's dependent variables of interest while also computing summary statistics (i.e., quantifying JEs, measuring the range of $p$-values and estimates across all models) for each dependent feature. Using Eq (10), for all vibrations, the impact of the presence of each adjuster on the value of the beta-coefficient of the independent primary variable of interest (referred to as $\beta_1$ in the above equations).

$$\text{absolute\_value}(\text{beta\_coefficient\_on\_primary\_variable})$$
$$\sim \beta_0 + \sum_{i=1}^{P} \beta_i \text{adjuster}_i + (1|\text{dependent\_variable}_{1..n}) \tag{10}$$

Each adjuster is encoded as a binary variable indicating its presence or absence in a given model. The random effect is used to account for cases when multiple dependent variables were analyzed. Using a random effect adjusts for variation in effect size due to specific dependent variables (e.g., if one is positively associated with an independent feature and one is negatively associated). Adjusting for these dependent variable–specific shifts enables this modeling strategy to identify underlying changes in effect size due to different independent adjusters and not dependent variables. In the event that the user specifies a single dependent variable (as in Eqs 3 to 9 and in the examples in our manuscript), a nonmixed model is fit. In either case, the interpretation of the beta coefficients on the adjuster variables is that in the presence of adjusters$_{1..P}$, the absolute value of the beta-coefficient on the primary variable increased/decreased, on average, by the values of $\beta_{1..P}$.

In the case that the user is only investigating one dependent variable at a time, the following, nonmixed model will be fit:

$$\text{absolute\_value}(\text{beta\_coefficient\_on\_primary\_variable}) \sim \beta_0 + \sum_{i=1}^{P} \beta_i \text{adjuster}_i \tag{11}$$

This is because, in this case, the random effect would only have 1 level (corresponding to the one dependent variable), rendering it meaningless. We investigated our dependent variables one at a time in this manuscript, so this is the model that was used to generate the results in Fig 4.

In addition to summarized "tidy" model statistics, we provide the full model output for Eqs 10/11 as part of quantvoe, enabling users to explore model features (e.g., residuals) independently.

### Package installation and testing

The package can be downloaded from GitHub. Detailed installation instructions and other package details are located in the GitHub README. Quantvoe is replete with an example

vignette to demonstrate use in the R terminal, unit testing suite, and example (bash) command line deployments.

## Datasets used in example

To demonstrate the utility of our tool, we computed VoE for a variety of clinical phenotypes (e.g., LDL cholesterol, triglycerides, glucose, vision). For all examples except 1, we used the 2005 to 2018 NHANES, which can be accessed in its raw components (including prescription data) at https://wwwn.cdc.gov/nchs/nhanes/ or, as we did, through the R package RNHANES (https://github.com/silentspringinstitute/RNHANES). This contains a total of 70,328 individuals, though the exact number of individuals differed slightly depending on the analysis in question (e.g., the dependent variable). For associations where BMI was the dependent variable, we used $N = 70,328$ individuals. For bone density, we used $N = 21,939$; for blood pressure and lisinopril, we used $N = 28,656$. For LDL, we used $N = 58,973$. For blood pressure and other indicators of heart disease, we used $N = 58,295$. For total cholesterol and other indicators of heart disease, we used $N = 58,973$. For vision, we used $N = 9,368$. For glucose, we used $N = 11,223$. Datasets used in analysis can be accessed on our GitHub repository.

Additionally, we computed VoE for the COVID-19 positivity outcome (testing positive versus negative) in 9,268 individuals, 7,724 controls, and 1,544 cases. The data from the UK Biobank are available upon application (https://www.ukbiobank.ac.uk/register-apply/).

## Computing total physical activity (TOTMETW)

We computed a series of variables to summarize the total minutes, vigorous and moderate minutes of activity per week from self-report physical activity variables. We used a metabolic equivalent of 8 for physical activity classified as vigorous, and a metabolic equivalent of 4 for moderate activities. The total metabolic equivalent for the week was the summation of vigorous, moderate, transportation, and leisure time activities.

## Downloading and processing NHANES and UK Biobank data

The scripts we used to minimally preprocess our data can be found at https://github.com/chiragjp/quantvoe/tree/main/manuscript/. Note that these are reliant on an existing dataframe containing NHANES prescription and processed physical activity (in units of METs) data and another dataframe containing UK Biobank COVID-19 test outcome, environmental, clinical, and sociodemographic data.

Generally speaking, for each association of interest, we created 2 RDS files: one containing the dependent variable(s) of interest (e.g., vision), the other containing the independent variables of interest. We (1) removed adjusters that were more than 50% missing data, (2) identified pairs of variables encoding redundant information and removed one (S3 Table), and (3) removed individuals with missing values for any of the independent variables of interest (i.e., primary variables). We additionally scaled and centered all numeric, continuous variables using R's scale() function with the default settings.

For the relationships queried in Fig 2, we examined the associations between overall vision (VIDLVA and VIDRVA) and carrot intake (FFQ0032), femur density (DXXOFBMD) and calcium intake (DR1TCALC), family monthly poverty index, the ratio of family monthly income to United States Health and Human Services poverty guidelines as a function of family size (INDFMMPI) and blood glucose (LBDGLUSI), and lisinopril usage (flag_LISINOPRIL) and average systolic blood pressure (mean of BPXSY1, BPXSY2, and BPXSY3). For the associations with lisinopril usage, we only used NHANES data between 2011 and 2018, as those years were the only times prescription usage was recorded. We indicated drug usage as 1 if a patient

indicated having taken the drug, and 0 as otherwise. Other variables (e.g., vision) were only recorded for a small set of years, resulting in our using a subset of the dataset in these cases. The carrot intake association was done using binary logistic regression. This is likely nonoptimal and was done for the sake of simplicity, given that this was meant to illustrate just an example of VoE. Another—more appropriate—approach for this would be to use an ordinal model.

We also examined associations between COVID-19 positivity and vitamin D levels using a subset of the UK Biobank cohort for which COVID-19 testing data was made available (as of July 17, 2020). The UK Biobank determines the COVID-19 test positivity outcome by microbiological (reverse transcriptase polymerase chain reaction [RT-PCR]) testing [55]. We define the outcome as the presence of at least one positive test result for a given participant. Adjusters in this dataset broadly include (a) clinical and diagnostic biomarkers of chronic disease and infection (e.g., white blood cell count, LDL cholesterol, BMI); (b) "environmental" factors (e.g., estimated nutrients consumed yesterday, infectious antigens, smoking history); (c) self-reported, doctor-diagnosed health and disease indicators (e.g., "diabetes diagnosed by doctor," "overall health rating," "vascular/heart problems diagnosed by doctor"); and (d) baseline socio-demographic factors (e.g., age, gender, average total household income after tax). Furthermore, since the UK Biobank measured clinical biomarkers during multiple visits, we computed the medians of biomarkers across visits. Additionally, we averaged quantitative environmental factors (which include the "estimated nutrients consumed yesterday" (23 exposures [e.g., estimated carbohydrate intake]) and "infectious antigens" (25 exposures [e.g., 1gG antigen for herpes simplex virus-1]) categories) over measurements from multiple visits. For categorical variables measured during multiple visits of a participant to the assessment center, we used the visit that contained the highest number of observations. We also performed rank-based inverse normal transformation (INT) of all real-valued adjuster variables to enhance comparability of associations across models. We performed the transformation using the RNOmni package (rankNorm function) [56] with the offset parameter set to 0.5 (as was suggested by Millard and colleagues [57]).

## Using quantvoe to model vibration of effects

For the cardiovascular associations, we queried the following dependent variables: body-mass-index (BMXBMI), total cholesterol (LBXTC), LDL-cholesterol (LBDLDL), and systolic blood pressure (mean of BPXSY1, BPXSY2, and BPXSY3). We queried the association between those 4 dependent variables and the following: caffeine intake (DR1TCAFF), sugar intake (DR1TSUGR), alcohol intake (DR1TALCO), smoking (LBXCOT), physical activity (TOTMETW), fiber intake (bDIETARY_FIBER_gm), total fat intake (DR1TFAT), total caloric intake (DR1TKCAL), and family history of coronary artery disease (MCQ300A) as well as all of the possible adjusters present. We additionally included BMI again as a 10th independent variable due to its role as both a risk factor and potential causative factor for heart disease [58], bringing the total list of associations we were studying to 39.

Given their reported ability to confound many associations and our aim to explore VoE beyond already established confounding, we included age and sex as baseline adjusters in all models fit [59],

For the vision/carrots analysis, we used logistic instead of linear regression, encoding the vision variable as 1 (20 in both eyes, as good as possible) or 0 (not equal to 20 in one or both eyes). Similarly, we used logistic regression for the COVID positivity analysis, where cases were encoded as 1, and controls as 0.

For the NHANES data, we additionally used their weighting schema, selecting the appropriate sample weights according to the dependent variable in question (e.g., WTMEC2YR for

exam data). We additionally used the SDMVPSU and SDMVSTRA columns to account for primary sampling units and strata, respectively.

For each association of interest, we used a maximum of 20 variables per model, 10,000 vibrations, and 1 core.

### Benchmarking

We set the FDR cutoff to 1 to avoid filtering any dependent variables. To avoid fitting $>2^{300}$ models, we ran the pipeline with a range of options for the max number of vibrations per feature 100, 250, 500, 1,000, 2,500, 5,000, 7,500, 10,000, 25,000, 50,000, 75,000, and 100,000. Similarly, to avoid fitting models with an excessive number of independent variables, we set the maximum number of variables in a model equal to 20. We again used weight and stratification variables as before and set the core usage to 1.

### Patient and public involvement

Patients or the public were not involved in the design, or conduct, or reporting, or dissemination plans of our research.

### Supporting information

**S1 Table. Glossary of terms.**
(XLSX)

**S2 Table. VoE output for Figs 2–4.**
(XLSX)

**S3 Table. Variables removed due to redundancy irrelevance from NHANES dataset.**
(XLSX)

**S4 Table. Software parameters and descriptions.**
(XLSX)

**S1 Fig. Regression validation plots for example confounder analyses.** Quadrant titles (e.g., Y ~ X) match the names in Fig 4.
(TIFF)

**S2 Fig. Software performance statistics for multiple dependent variables.** In terms of (A) runtime for different numbers of vibrations, (B) memory usage for different numbers of vibrations. Data underlying these plots are available at https://figshare.com/account/home#/projects/120969.
(TIFF)

### Acknowledgments

We thank Harvard Research Computing for providing compute resources for this work.

### Author Contributions

**Conceptualization:** Braden T. Tierney, Aleksandar D. Kostic, Arjun K. Manrai, Chirag J. Patel.

**Data curation:** Braden T. Tierney, Elizabeth Anderson, Kajal Claypool, Sivateja Tangirala.

**Formal analysis:** Braden T. Tierney.

**Funding acquisition:** Arjun K. Manrai, Chirag J. Patel.

**Investigation:** Braden T. Tierney, Chirag J. Patel.

**Methodology:** Braden T. Tierney, Elizabeth Anderson, Yingxuan Tan, Chirag J. Patel.

**Project administration:** Braden T. Tierney, Aleksandar D. Kostic, Arjun K. Manrai, Chirag J. Patel.

**Resources:** Braden T. Tierney.

**Software:** Braden T. Tierney, Elizabeth Anderson, Yingxuan Tan.

**Supervision:** Aleksandar D. Kostic, Arjun K. Manrai, Chirag J. Patel.

**Visualization:** Braden T. Tierney.

**Writing – original draft:** Braden T. Tierney, Kajal Claypool, Sivateja Tangirala, Aleksandar D. Kostic, Arjun K. Manrai, Chirag J. Patel.

**Writing – review & editing:** Braden T. Tierney, Yingxuan Tan, Aleksandar D. Kostic, Arjun K. Manrai, Chirag J. Patel.

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
