## [Editor Report · Decision Letter 0]

14 Apr 2021

Dear Dr Patel, 

Thank you for submitting your revised manuscript entitled "Charting a course for meaningful discovery in biomedical data science" for consideration as a Meta-Research Article by PLOS Biology.

Your revisions have now been evaluated by the PLOS Biology editorial staff, as well as by the Academic Editor, and I'm writing to let you know that we would like to send your submission out for external peer review.

Please re-submit your manuscript within two working days, i.e. by Apr 16 2021 11:59PM.

Kind regards,

Roli Roberts

Senior Editor

PLOS Biology

---

## [Decision Letter · Decision Letter 1]

9 Jun 2021

Dear Dr Patel,

Thank you very much for submitting your manuscript "Charting a course for robust discovery in biomedical data science" for consideration as a Meta-Research Article at PLOS Biology. Your manuscript has been evaluated by the PLOS Biology editors, an Academic Editor with relevant expertise, and by three independent reviewers.

You'll see that all three reviewers are broadly fairly positive about your study, but reviewers #1 and #2 have overlapping concerns about the novelty of your approach, and ask for clarification as to how you expect people to use quantvoe. Reviewer #3 also shares some of these concerns, and also seems to ask about the potential pitfalls of running every permutation (the brute force approach) without careful consideration of what are reasonable variables to include, what the underlying (implicit) causal model is, etc.

In light of the reviews (below), we will not be able to accept the current version of the manuscript, but we would welcome re-submission of a much-revised version that takes into account the reviewers' comments. We cannot make any decision about publication until we have seen the revised manuscript and your response to the reviewers' comments. Your revised manuscript is also likely to be sent for further evaluation by the reviewers.

We expect to receive your revised manuscript within 3 months. 

**IMPORTANT - SUBMITTING YOUR REVISION**

*Re-submission Checklist*

*Published Peer Review*

*PLOS Data Policy*

*Blot and Gel Data Policy*

Sincerely,

Roli Roberts

Roland Roberts

Senior Editor

PLOS Biology

rroberts@plos.org

REVIEWERS' COMMENTS:

Reviewer #1:

[identifies himself as Robert E. Arbon]

 This paper presents a tool for performing a type of multiverse analysis known as a vibration of effects (VoE). This tool is applicable in the situation where one is interested in modelling the effect of X on Y but where there are a large number of potential confounders and no underlying theory to inform the modelling strategy. The tool, quantvoe, aims at "automatically identifying what measured adjusting variables drive inconsistency in associations between X and Y". It does this by modelling a wide range of model specifications and then modelling the impact of adjustments on the size of a measured association using a linear (potentially multilevel) model. 

As already noted in the editor's comments, the approach of VoE and multiverse analysis is not novel but developing a robust tool to perform this analysis is worthy contribution the field. The approach to identifying variables which affect associations is an interesting and novel one. 

The significance of the work is hard to judge (see revisions below) and the authors need to make it very clear how this tool is to be used in a practical research sense. 

I recommend this paper for publication, once a number of major revisions have been made. 

Major revisions: 

1. The title of the paper does not match the content and one of them should change (preferably the content). It is not clear how to use the output of the quantvoe to inform further research. You mention in passing (71 - 73) using VoE to prioritise correlations, please expand on this, or develop the examples given in the same manner, even if they are hypothetical situations. A figure summarising quantvoe's place in the broader research workflow would be an excellent addition but not necessary. 

2. In the same vein as point 1, please provide some indication as to what the Janus effect actually means in practice. 

3. I was not able to follow the instructions for the tests of quantvoe (on the Github repo) as the necessary datasets were not provided. Please provide this so this software can be tested. Alternatively, please provide detailed instructions/commands to run with the NHANES dataset provided. 

4. Please discuss alternative approaches to biomedical research in the same context i.e., when there is little or no theory and lots of measured variables. There are other types of sensitivity analysis e.g., VanderWeele's E values. Please compare these approaches to quantvoe using the examples given. 

5. Please provide a justification for your claim: 

…our package circumvents the need for a priori selection of a small set of candidate variables…

6. Please provide validation of the modelling approach for identifying important adjusting variables, i.e., equations 10 and 11 for some of the examples used. E.g., residual plots etc. 

7. There are also many different sources bias which have not been mentioned in the caveats section: quantvoe's treatment of missing variables, the effect of non-linearities, lack of account of global model fit (e.g., R2 or AIC for each vibration). The effect of non-linear terms in particular, as this will cause the number of models to increase drastically. 

8. Please remove 'madman' from your Github readme which is linked to from the paper as it's ableist language. Unless this has to do with the metaanalysis of gene expression datasets, in which case please clarify. 

There are also some minor points which would aid the readability: 

1. Please point the reader to the full definitions of the Janus effect and VoE, when they first appear. e.g., VoE is rigorously defined in lines 385 - 387, so please link to this definition. 

2. The type setting of the mathematical equations is difficult to read. Please consider using standard mathematical/statistical notation, or at least changing the font from italic to regular. 

3. The font size of the charts is also very hard to read, please make as large as possible. 

4. Make sure all acronyms are defined, e.g., TOTMETW.

5. Grammar errors: lines 55, 62. 

6. Inappropriate use of binary logistic regression for the vision/carrots analysis, it should be an ordinal model. No need to re-run as this was an illustrative example, but please note the reason for doing this, e.g., speed or simplicity. 

Reviewer #2:

[identifies himself as Florian Naudet]

This is an important manuscript for researchers involved in the field of clinical epidemiology. This is a meta-research paper that focuses on the on of the 5 important areas of meta-research, i.e. "methods".

I support its publication, providing authors can address the following comments :

GENERAL COMMENT

- I share the initial concern of the editor that the paper was not so novel. There are previous papers dealing with this concept of VoE (by the authors of this manuscript who were among the fist to identify it in epidemiology) and by other in different scientific fields (e.g. psychology etc.). However, I also appreciate that the authors are introducing a new R library and illustrate its use with some examples. I therefore consider that it has some interest/novelty. Still the decision concerning this point (novelty) can only be an editorial decision. Should the editor of PLOS Biology find it not so novel, I suggest forwarding it to another journal (e.g. PLOS MEDICINE or PLOS ONE). 

INTRODCUTION

- I also thank the authors for providing a clear introduction to the concepts they are studying. The text is quite simple to read, there is no specific jargon and I think that it can be suitable for a large audience. For instance, the authors provide a series of examples were VoE can be at play. In their very engaging introduction, however authors refer to a paper (l.38) about "chocolate consumption and the probability of winning a Nobel Prize". Of course this paper illustrates the problems of nutritional research that can sometimes find very spurious associations. Still, I think this paper was not the best choice as it was not a very serious paper and, in my understanding, it was a paper written in a tongue in cheek way to point problems arising from ecological fallacy. I therefore suggest the use of another example. 

- Although clear, the introduction is long and sometimes I found it a little bit repetitive and even contradictory :

. e.g. "But few of the many possible types of multiverse analysis have been fully automated and many require manual configuration." (l.67) ; 

. and (l.75) "There is a need for tools to operationalize multiverse analysis through comparing model adjusting strategies."

. and (l.83) "However, unlike other disciplines, no standardized software package exists to systematically model VoE in massive datasets with hundreds of potential adjusting variables"

- > I therefore suggest to shorten the introduction and to clearly provide a short but exhaustive description of all tools that exist to explore VoE, by discipline and a description of the specific lack of tools in clinical epidemiology and highlight the interest of their package. I just mean that the introduction can still be a little bit simplified and more focused on the use of package among all existing initiatives in this area. 

- l.52 "diet" appears twice in "(e.g. age, exercise, diet, diet)".

RESULTS / DISCUSSION

- I found this part very clear ; 

- I suggest that the term Janus effect (first appearance l.121 if I'm correct) can be introduced before (e.g. in the introduction) / I understand that it is detailed in the methods section but as the methods are reported at the end of the paper, it is better to introduced it before ; 

- In this part I would be interested to know the certainty we have a priori in the various associations explored : 

. Some have an obvious causal association (i.e. anti-hypertensive agents lower blood pressure) ; 

. Other are somewhat unlikely (e.g. Vitamin D levels affect covid, carrot etc) ; 

-> I would be interested in a more systematic assessment of these priors that can help interpreting their results (authors tackle this but don't address it full l.216). 

- In the same spirit, I would be interested, for each association, to see the possible confounders that can be at play in the data sets that were considered (e.g. indication bias for anti-hypertensive agents). 

- As part of the question about novelty, I found it very innovative and interesting to analyse and detail the impact of each variable that could be entered in the model. I think that it point is not put forward in this paper. Still, it is very important in my opinion. 

- (l.209, l.210, l.211) this part of the discussion can be expanded to detail the differences between the other methods and this new one with their possible strengths and weaknesses.

- There is an important interest in VoE currently. This is an excellent reference and I think that the authors missed it : https://royalsocietypublishing.org/doi/pdf/10.1098/rsos.201925 / I suggest using it and discussion in depth that model specification is only one aspect of VoE ; 

- I really appreciate that the authors used various examples in this paper using different databases and research question. To adopt a new tool and a new procedure, one need various examples and this is a good point here. I would however like to see more discussion about this : how many case studies may be necessary before adopting it ;

- And importantly I would expect more discussion on "How can one use this framework in a practice ?" If it was badly used, this extreme form of sensitivity analysis could be a massive destruction weapon for the field of epidemiology. If correctly and smartly used, I'm sure that it can increase value of the field. Can authors elaborate a little bit on this point ? 

METHODS 

This is very clear. Only two points : 

- Please specify if this study was registered or not (sorry if I missed it). I always think that pre registration is a good thing, even for this kind of methodological research. If it was not pre-registered, I would like to see this explicitly acknowledged and to read the reasons for this. 

- Last, I think that some points can be moved in the web appendix (e.g. instructions to download the package) but this is rather a suggestion. 

Reviewer #3:

Charting a course for robust discovery in biomedical data science

This paper restate and reinforce the communication of the so called "vibration effect", "multiverse", "Analytical flexibility", etc in a general context. 

The term vibration of effect should be clearly positioned compared to previous references.

The brute force approach described should be clearly related to variable selection, variable importance, and multiple comparison statistical techniques in the introduction. If scientists are to use the package to decide on the presence of an association, the reporting should consider the factors influencing the set of results. In particular, improvement or loss of measured association should be predictable through estimation of the linear dependency of confounders with the dependent variable. 

As noted by the authors, the number of dependent variables can be high and strategies to account for these cases should be better described. 

It seems to this reviewer that the key aspect is wether reporting the number of positive association compared to negative association is reasonable in general (Janus effect). Consider the case of one confounder is highly correlated with the independent variable, this would lead to 2^(p-1) models with very poor association (p is the number of dependent variables), this number would rapidely increase with the number of covariate related to the independent variable. However, this would assume that it makes sense to include these covariates, which is a strong assumption of the proposed framework. The context in which this type of brute force analysis make sense should be therefore be emphasized in the introduction as well as in the discussion, since linear models are generally interpreted as causal models.

Implementation: the package seems to be lacking a unit testing suite.

---

## [Decision Letter · Decision Letter 2]

20 Aug 2021

Dear Dr Patel,

Thank you for submitting your revised Meta-Research Article entitled "Charting a course for robust discovery in biomedical data science" for publication in PLOS Biology. I have now obtained advice from the original reviewers and have discussed their comments with the Academic Editor. 

Based on the reviews, we will probably accept this manuscript for publication, provided you satisfactorily address the remaining points raised by the reviewers. Please also make sure to address the following data and other policy-related requests.

We suggest that you change the title to make it more informative and explicit. It should include the phrase "Vibration of effects analysis" and should also cover what type of study you consider. You are welcome to email me your suggestion before resubmitting to discuss it.

DATA POLICY:

Regardless of the method selected, please ensure that you provide the individual numerical values that underlie the summary data displayed in the following figure panels as they are essential for readers to assess your analysis and to reproduce it: Figure 4ACEG.

**Please also ensure that figure legends in your manuscript include information on where the underlying data can be found, and ensure your supplemental data file/s has a legend.**

We expect to receive your revised manuscript within two weeks.

*Published Peer Review History*

*Early Version*

Sincerely,

Senior Editor,

rroberts@plos.org,

PLOS Biology

Reviewer remarks:

Reviewer #1: Robert Edward Arbon

Reviewer #2: Florian NAUDET

Reviewer #3: Anonymous

Reviewer #1: This manuscript is much improved as is the Github repository. There are still some issues with the software but as these might be system dependent I will move these comments to issues on Github. The README is clear and comprehensive, indeed, one of the best I have seen. I was able to run the 'beginner' example with no problems. There are still grammatical type setting issues which I have sent separately in a marked up word document to the editor.

I recommend this for publication with the following minor revision: re: Janus effect - some reference to the literature on sign changes in regression modelling would be beneficial, e.g., doi: 10.1186/1742-7622-5-5

Reviewer #2: I would like to thank the authors for revising their paper. I think that the paper is in a better shape now and I recommend it for publication.

Reviewer #3: The authors have well addressed the points raised in the previous review and the relation of the proposed technique with others is now well described.

One remaining point:

I am a little confused by the use of the terms "highly discordant" in the abstract :

77.3% [associations] were highly discordant, with greater than 5%"5% of all models generating conflicting (i.e. positive vs. negative) results"

Given the possible correlations between model variables, one would actually expect some negative associations.

I would therefore specify that this is not an unexpected result and only indicates the importance of the model selection, and reinforce the importance of the use of classical model selection techniques in the discussion.

---

## [Editor Report · Decision Letter 3]

24 Aug 2021

Dear Chirag,

On behalf of my colleagues and the Academic Editor, Marcus Munafo, I'm pleased to say that we can in principle offer to publish your Meta-Research Article "Leveraging vibration of effects analysis for robust discovery in observational biomedical data science" in PLOS Biology, provided you address any remaining formatting and reporting issues. These will be detailed in an email that will follow this letter and that you will usually receive within 2-3 business days, during which time no action is required from you. Please note that we will not be able to formally accept your manuscript and schedule it for publication until you have made the required changes.

IMPORTANT: I note that, due to a misunderstanding, you have renamed your two Supplementary Figures and four Supplementary Tables as six Supplementary Data files. This is not appropriate (the Supp Data files should be reserved for raw data and data that underlie the Figs, which you have deposited in Figshare), and should be reverted, both in the file names, in their legends, and in the in-text call-outs to them. Our colleagues that handle the next few stages have been instructed to request this change.

PRESS: We frequently collaborate with press offices. If your institution or institutions have a press office, please notify them about your upcoming paper at this point, to enable them to help maximise its impact. If the press office is planning to promote your findings, we would be grateful if they could coordinate with biologypress@plos.org. If you have not yet opted out of the early version process, we ask that you notify us immediately of any press plans so that we may do so on your behalf.

Best wishes,

Roli 

Roland G Roberts, PhD 

Senior Editor 

PLOS Biology

rroberts@plos.org